# Characterization of Molting Process during the Different Developmental Stages of the Diamondback Moth *Plutella xylostella*

**DOI:** 10.3390/insects13030289

**Published:** 2022-03-15

**Authors:** Licheng Gu, Zhiwei Wu, Xiaotong Wu, Yuenan Zhou, Pei Yang, Xiqian Ye, Min Shi, Jianhua Huang, Xuexin Chen

**Affiliations:** 1Institute of Insect Science, College of Agriculture and Biotechnology, Zhejiang University, Hangzhou 310058, China; 11916098@zju.edu.cn (L.G.); wuzhiweistudy@163.com (Z.W.); xtongwu@zju.edu.cn (X.W.); ynzhou@zju.edu.cn (Y.Z.); yangpei_1104@163.com (P.Y.); yxqfi@hotmail.com (X.Y.); shimin0623@zju.edu.cn (M.S.); jhhuang@zju.edu.cn (J.H.); 2Guangdong Laboratory for Lingnan Modern Agriculture, Guangzhou 510642, China; 3Ministry of Agriculture Key Laboratory of Molecular Biology of Crop Pathogens and Insect Pests, Zhejiang University, Hangzhou 310058, China; 4Key Laboratory of Biology of Crop Pathogens and Insects of Zhejiang Province, Zhejiang University, Hangzhou 310058, China; 5State Key Laboratory of Rice Biology, Zhejiang University, Hangzhou 310058, China

**Keywords:** *Plutella xylostella*, behavioral sequence, molting, ecdysis, morphological characters

## Abstract

**Simple Summary:**

The diamondback moth, *Plutella xylostella* (L.) is the most important pests of cruciferous crops worldwide. Although there are numerous studies on the general life cycle of *P. xylostella*, the detailed descriptions of the morphological transformation and behavioral sequence during molting are rarely provided and visualized. In this paper, we provided the duration and photographic details of staging criteria of each stage of egg hatching, larval–larval ecdysis, larval–pupal metamorphosis and adult eclosion, and post-eclosion behavior of *P. xylostella*. Several new characters in the molting process that were previously not described in other lepidopteran insects were found, i.e., the larvae contracted anterior-posteriorly then dorsal-ventrally during pre-ecdysis, and the antennae waved backward then forward in the post-eclosion behavior. Our findings add an important piece of knowledge about the molting biology of lepidopteran insects.

**Abstract:**

The molting process of the lepidopteran insects is observed for many species. However, the detailed description of the morphological transformation and behavioral sequence during molting are rarely provided and visualized. Here, we described the molting process of the diamondback moth *Plutella xylostella* by providing the duration and photographic details of staging criteria of each stage using stereo microscopy and a digital video camera. We divided the morphological transformation of egg development and hatching into five stages, the larval–larval ecdysis and the larval–pupal metamorphosis into five stages, the pupal development and eclosion into three stages, and the post-eclosion behavior into four stages. Several new characters in the molting process that were not previously described in other lepidopteran insects were found, i.e., the larvae contracted anterior-posteriorly then dorsal-ventrally during pre-ecdysis, and the antennae waved backward then forward in the post-eclosion behavior. Our findings will deepen the knowledge of the molting biology of lepidopteran insects and facilitate the study of the underlying mechanisms.

## 1. Introduction

Insects produce a new exoskeleton periodically, which is defined as molting. When a new exoskeleton is produced, the old one must be shed afterward, which is termed ecdysis [1,2].

As model insects, species of *Manduca* and *Drosophila* have received the most attention in this regard. The molting process of *Manduca sexta* [3,4,5] and *Bombyx mori* [6], two lepidopteran model insects, have been intensively studied in different developmental stages, including larval–larval (L-L) ecdysis, larval–pupal (L-P) metamorphosis, and adult eclosion. Several studies also have described the molting process in other non-model lepidopteran insects, such as the L-P metamorphosis of *Heliothis virescens* [7], the L-L ecdysis and L-P metamorphosis of *Pseudoplusia includens* [8] and *Trichoplusia ni* [9], the pupal development of *Antheraea pernyi* [10], the eclosion of *Hyalophora cecropia* [11] and *Dioryctria abietella* [12], and the post-eclosion behavior of *Automeris aurantiaca* [13], *Dioryctria abietella* [12], and *Heliothis zea* [14]. Their molting processes have been described by the presence of morphological characters and behavioral sequences, though with minor differences. The L-L ecdysis of lepidopteran insects is composed of the head capsule slip (HCS), the pigmentation of mandibles, and the behaviors named pre-ecdysis and ecdysis. The L-P metamorphosis of lepidopteran insects consists of wandering, cocooning, HCS, shrinkage and pigmentation of the cuticle, pre-ecdysis, and ecdysis. The eclosion consists of pre-eclosion and eclosion behavior, followed by the wing expansion termed post-eclosion behavior.

The molting process of another model insect, the dipteran insect *Drosophila melanogaster*, during the larval stages has been characterized by double mouth hooks (dMH), double vertical plates (dVP), trachea collapse (TC), air filling the space between the old and new intimae (AF), anterior-posterior contractions (AP), a squeezing wave of dorsal-ventral contraction (SW), and, finally, forward escape [15]. The L-P metamorphosis of *D. melanogaster* is composed of 24 stages with 51 visible changes from the wandering third-instar larva to cryptocephalic pupa to phanerocephalic pupa [16]. The morphological characters during the eclosion have been depicted according to the resorption of molting fluid and air filling [17], and the following behavior sequences are pre-eclosion, which comprises of head inflation, head tracheal filling, and ptilinium extension, followed by quiescence and eclosion, which is composed of head inflation, head thrusts, contractions of the thorax, and peristaltic abdominal contractions [18].

The molting process has been described in other holometabolous insects, including *Aedes aegypti* [19], *Apis mellifera* and *Tenebrio molitor* [20], and *Tribolium castaneum* [21], and the final molt in some hemimetabolous insects, i.e., *Anax imperator* [22], *Pantala flavescens* [23], *Rhodnius prolixus* [24], *Teleogryllus oceanicus* [25,26,27], and *Carausius morosus* [28]. 

The morphological characters in egg development have been described in the dipteran insects *Chrysomya megacephala* [29] and *Hermetia illucens* [30] and the hemipteran insect *Empoasca onukii* [31]. In general, it consists of the chorion becoming dented and shriveled, embryo stemmata pigmentation, and segmentation. The following larval hatching is described in *M. sexta* [32].

Characteristics for staging the molting process have been developed for model insects, especially in *M. sexta* and *B. mori*, two insects with large body sizes. Yet, there is only scattered information available for non-model species. Detailed descriptions of the morphological transformation and behavioral sequence during each molting process in the full life cycle of non-model insects, especially small ones, with duration and photographs, are rarely provided.

The diamondback moth, *Plutella xylostella* (L.) (Lepidopteran: Plutellidae) is the most important pest of cruciferous crops worldwide [33,34,35]. The life cycle of *P. xylostella* has been observed [36], but the morphological changes and behavioral sequences of the molting process remained undescribed. 

In this study, we observed the morphological changes and behavioral sequences of the molting process of *P. xylostella* during different stages, including egg hatching, L-L ecdysis, L-P metamorphosis, pupa-adult eclosion, and post-eclosion. Detailed descriptions with color photos are provided for the first time. The information of the molting process would be useful to properly stage this insect and analyze in detail the phenotypes and effects of insecticides.

## 2. Materials and Methods

### 2.1. Maintenance of P. xylostella

The *P. xylostella* used in the study were collected in Hangzhou, Zhejiang province, China (30.3009° N, 120.0870° E) and had been continuously maintained for over 100 generations. Both the larvae and adult of *P. xylostella* were maintained on a 14 h light: 10 h dark cycle at 25 ± 0.3 °C and 50 ± 10% relative humidity. Unless described otherwise, all observations were made at the condition mentioned above. The larvae of *P. xylostella* were reared on cabbage, and the adults of *P. xylostella* were fed a 10% (*w*/*v*) honey solution in 30 cm × 30 cm × 30 cm rearing cages.

### 2.2. Observation and Staging of the Molting Process

#### 2.2.1. Egg Hatching

A piece of pan paper with cabbage juice inside the rearing cage was used to collect eggs. The pan paper was removed after 10 min and the eggs were immediately washed off with ultrapure water. Then, those eggs were arranged with a brush in a 9 mm petri dish. The morphological characteristics of the eggs were observed under a ZEISS Stemi 305 compact stereo microscope (Zeiss, Oberkochen, Germany) every 1 h and every 20 min at around 12 h before the eggs hatched. Videos were taken about 2 h before the eggs hatched. Thirty-three eggs were observed.

#### 2.2.2. Larval–Larval and Larval–Pupal Ecdysis

The L-L and L-P molting processes were staged referring to the morphological characters of *H. virescens* [7], *P. includens* [8] and *T. ni* [9] and to the pre-ecdysis and ecdysis behavioral sequences of *M. sexta* [37,38,39]. To observe the molting process of *P. xylostella*, larvae were transferred to a petri dish with fresh cabbage leaves. All the larvae that were in the molting process were removed before observation. Larvae that subsequently ceased feeding and migrated to the lid of the petri dish were placed in a new petri dish individually and the time of moving to the lid was recorded as the beginning of their molting process. The morphological characteristics of over 30 larvae were observed and recorded under a ZEISS Stemi 305 compact stereo microscope (Zeiss, Oberkochen, Germany) every 10 min until the ecdysis process was complete. Then, the pupae were kept for the observation of the L-P metamorphosis.

#### 2.2.3. Eclosion and Post-Eclosion Behavior

The morphological characteristics of the development of pupae were staged using the criteria of *B. mori* and *M. sexta* [40] and *A. pernyi* [10]. The development of pupae was recorded with a digital video camera. The behavioral sequences of eclosion were staged using the criteria of *B. mori* [6] and *M. sexta* [41], and the post-eclosion behavioral sequences were staged using the criteria of *M. sexta* [3,4,5], *H. zea* [14], and *D. abietella* [12]. Thirty-six pupae were observed.

All above observations were carried out at a room temperature of 25 ± 1 °C. The observation of morphological characteristics was conducted by direct observation and with a ZEISS Stemi 305 compact stereo microscope (Zeiss, Oberkochen, Germany). 

### 2.3. Photography and Videography

All pictures were taken with a Keyence VHX-7000C automated imaging system (Keyence, Osaka, Japan). A Sony FDR-AX100 digital video camera (Sony, Tokyo, Japan) and a SOPTOP DMSZ7 stereo microscope (Sunny Optical Technology (Group) CO., Ltd., Zhejiang, China) were used to record the behavioral sequences of egg hatching, L-L ecdysis, L-P metamorphosis, adult eclosion, and post-eclosion.

### 2.4. Statistics

All data were analyzed using Microsoft Office Excel 2013 software. The data are presented as means ± s.e.m.

## 3. Results

*P. xylostella* undergoes six phases of molting in one life cycle, i.e., the egg hatching, the first instar larva-second instar larva (L1–L2), second instar larva-third instar larva (L2–L3), third instar larva-fourth instar larva (L3–L4) L-L ecdysis, the L-P metamorphosis, and the adult eclosion. The duration of each stage is provided in Table 1.

### 3.1. The Egg Development and Hatching Process

In stage I of egg development, the egg was evenly white, and the chorion was full when it was just laid. Then, the substance in the egg condensed, and the chorion was dented and shriveled (Figure 1A, lasted 36 ± 0.2 h). In stage II, pigmentation of the stemmata began. A red spot was visible on the egg. It darkened and finally turned black (Figure 1B, lasted 27 ± 0.3 h). Stage III was the head capsule pigmentation. The head and mandibles of the embryo moved sporadically. At this stage, the segmentation of the thorax and abdomen was complete and the setae were visible (Figure 1C, lasted 3 ± 0.1 h). In stage IV, the cuticular structures of the embryo have acquired their distinctive larval morphology and the liquid in the eggs was fully absorbed. The chorion shrank and clung to the surface of the embryo (Figure 1D, lasted 3 ± 0.2 h). In the hatching process, the newly emerged first instar larva waved its head to find a breaking point and gnawed a hole in the chorion to permit its passage. After the hole was enlarged sufficiently, the head of larva emerged foremost, uncoiling the remainder of the body slowly as it moved out of the chorion. (Appendix A, lasted 0.2 ± 0.02 h). The following observations showed that no feeding on the chorion occurred after the hatching process was complete. The egg development and hatching process lasted 69 ± 0.3 h in average (Table 1).

### 3.2. The Larval–Larval Ecdysis

The L-L ecdysis is virtually the same in all instars. In stage I, the larva ceased feeding, migrated to the dish cover, and spun a few strands of silk to form a thin silk pad or sometimes a “cocoon” which prevented them from falling to the ground. There was no significant change on its head capsule in stage I (Figure 2A, lasted 91 ± 4 min for L1–L2, 86 ± 4 min for L2–L3, and 90 ± 4 min for L3–L4 ecdysis). Stage II was marked by the head capsule slip (HCS). Six stemmata of the old head capsule turned white, which meant stemmata pigment spots separated from the old head capsule. The old head capsule gradually slipped anteriorly and passed all stemmata pigment spots until all spots emerged from the posterior margin (Figure 2B, lasted 72 ± 3 min for L1–L2, 82 ± 2 min for L2–L3, and 96 ± 2 min for L3–L4 ecdysis). In the following stage III, the larvae kept relative quiescence. All stemmata on the new head capsule had already emerged from the posterior side of the old head capsule in this stage (Figure 2C, lasted 378 ± 6 min for L1–L2, 342 ± 4 min for L2–L3, and 438 ± 4 min of L3–L4 ecdysis). Then, in stage IV, the mandibles of the pharate larva were pigmented and gradually became visible through the old head capsules (Figure 2D). After the full pigmentation of the mandibles, a movement named pre-ecdysis behavior I first appeared as isolated anterior-posterior rhythmic contractions that gradually increased in frequency (Appendix A). Then, the pre-ecdysis behavior II, i.e., dorsal-ventral contractions, gradually started with anterior-posterior rhythmic contractions that continued simultaneously. Then, the anterior-posterior contractions gradually weakened and, finally, terminated while the dorsal-ventral contractions increased in frequency. The abdominal prolegs retracted and extended in some segments during the latter portion of pre-ecdysis II, and they were not coordinated with the contraction of the abdomen. The prolegs of the larvae could not stay with the surface beneath the abdomen in some cases, which meant the larvae may “lie down” or fall from a higher place without the protection of the “cocoon” (Appendix A). The pigmentation of mandibles, in pre-ecdysis I and II, which were defined as stage IV, lasted 64 ± 3 min in L1–L2, 76 ± 2 min in L2–L3, and 114 ± 3 min in L3–L4 ecdysis. Finally, in stage V, ecdysis behavior started. Waves of anterior-directed peristalses propelled the old cuticle backward until it eventually split at the dorsal ecdysial suture in the thorax. It was then pushed down and away from the body (Appendix A). The ecdysis behavior lasted 3 ± 0.1 min in L1–L2, 3 ± 0.1 min in L2–L3, and 4 ± 0.4 min in L3–L4 ecdysis. In conclusion, it lasted 605 ± 7 min of L1–L2, 585 ± 5 min of L2–L3, and 736 ± 5 min of L3–L4 larval–larval ecdysis in total. 

### 3.3. The Larval–Pupal Metamorphosis

The L-P metamorphosis began with wandering behavior. In stage I, the larva ceased feeding and wandered for a pupation place with sporadic spinning behavior. When the pupation place was located (mostly on the dish cover), the larva spun a few strands of silk beneath its body and kept relative quiescence thereafter. Stage I lasted 172 ± 14 min. Stage II was cocooning behavior. There was a white bar among the stemmata on the head capsule of fourth instar larvae. There were no significant changes in the stemmata and the white bar in the stage II (Figure 3A, lasted 313 ± 11 min). The larva started spinning again and was gradually enclosed in a thin silk cocoon, with its segments becoming shortened during the cocooning (Figure 3D and Appendix A). The larva kept ventral-side-down until the end. Then, the stemmata pigment spots detached from the head capsule and faded. In stage III, the head capsule slipped anteriorly and ventrally. The white bar kept its position among the stemmata. The stemmata pigment spots and the white bar progressively faded at this stage (Figure 3B, lasted 217 ± 9 min). During the HCS process, the thorax bulged ventrally; thus, the head and thorax of the prepupa started bending and finally stayed at an obtuse angle. In the following stage IV, the stemmata pigments migrated away from the posterior margin of the head capsule, merged as one pigment spot, and distinctly faded, while the white bar was left inside the head capsule and almost vanished (Figure 3C). Stage III was further highlighted by a morphological marker that wings, antennae, and the siphoning mouthpart of the future adult beneath the cuticle of the prepupa had already emerged and did not cling to the body (Appendix A). At the beginning of stage IV, the prepupa kept relative quiescence until pre-ecdysis behavior I and II took place. Like the pre-ecdysis I and II in L-L ecdysis, the rhythmic anterior-posterior contractions of pre-ecdysis I were weak in the beginning and gradually strengthened. (Figure 3E, and Appendix A). As the rhythmic anterior-posterior contractions continued, the rhythmic dorsal-ventral contractions of pre-ecdysis II gradually started. Both behaviors took place simultaneously, then the anterior-posterior contractions of pre-ecdysis I gradually weakened and, finally, terminated. In the latter portion of stage IV, only the dorsal-ventral contractions continued and gradually strengthened in the anterior abdominal region, especially at the first and second segment of the abdomen, which produced the subtle “bowing” movements of the head and the thorax (Figure 3F and Appendix A). The retraction and extension of the abdominal prolegs were not observed during this stage compared to the L-L ecdysis. Stage IV lasted 459 ± 5 min in total. The following stage V, ecdysis behavior, was also similar to that in L-L ecdysis, which lasted 4 ± 0.4 min. In stage V, waves of anterior-directed peristalses, which started from the last segment, propelled the old cuticle backward until it eventually split at the dorsal ecdysial suture in the thorax. The exuviae were then pushed down and away from the body (Appendix A). The whole larval–pupal metamorphosis lasted 1161 ± 22 min. 

### 3.4. The Pupal Development and Adult Eclosion

The entire process of pupal development is defined as stage I. The pupa was evenly green at first, with sporadic abdomen spinning (Figure 4A, lasted about 6 h). Then, the compound eyes, antennae, and appendages turned yellow distally (Figure 4B, lasted about 66 h). Next, the compound eyes turned black. Meanwhile, the antennae, appendages, wings, and thorax turned yellow. The last two segments of the abdomen turned yellow with other segments kept green (Figure 4C, lasted about 12 h). Thereafter, the antennae, appendages, wings, and thorax turned black. The last two segments of the abdomen turned brown and the other segments turned yellow (Figure 4D, lasted about 6 h). At the end of stage I, the pre-eclosion behavior gradually took place as rhythmic dorsal-ventral contractions (Appendix A). Stage I lasted 90 ± 1 h. Stage II of the intersegmental membrane of the pupa inflated with the metallic reflection on the scale of the pharate adult because of the full absorption of the molting fluid (Figure 4E and Appendix A, lasted 62 ± 1 min). At last, stage III, the eclosion behavior lasted 3 ± 0.2 min. The pupal development and adult eclosion of *P. xylostella* lasted 91 ± 1 h in total (Table 1). Waves of anterior-directed peristalses, which started from the last segment, propelled the old cuticle backward with a violent “shrugging” of the wing bases until it eventually split at the dorsal ecdysial suture in the thorax. The exuviae were then pushed down and away from the body (Appendix A). The newly emerged adults did not break the cocoon during eclosion since there are two “eclosion holes” on the anterior and posterior sides of the cocoons (Appendix A).

### 3.5. The Post-Eclosion Behavior of P. xylostella Moth

In stage I, the newly molted adult remained stationary and antennae waved forward as soon as the adult emerged from its pupa. Later, the adult started crawling to find a place for wing expansion (Figure 5A, lasted 3 ± 0.2 min). In stage II, the wings first elongated backward horizontally (Appendix A), then erected vertically as elongation continued (Figure 5B and Appendix A, lasted 7 ± 0.2 min). In the following stage III, the wings folded downward and covered the dorsum of the adult moth. Meanwhile, the antennae suddenly waved backward (Figure 5C, lasted about 7 ± 0.3 min). In stage IV, the antennae gradually waved forward again. At the end of this stage, the angle between the two antennae was far less than stage I (Figure 5D and Appendix A, lasted 10 ± 0.6 min) (Table 1).

## 4. Discussion

The molting process of the diamondback moth *P. xylostella* appeared similar to the stereotypic morphological characteristics and behavioral sequences that have been described and quantified in a number of lepidopteran insects, such as *M. sexta* [37,38,39,41,42], *T. ni* [9], *H. virescens* [7], *P. includens* [8], and *B. mori* [6]. Most of the morphological characteristics and behavioral sequences during the full-life molting processes of *P. xylostella* described here are similar in general to those mentioned above, while some are different and some had not been reported before, as discussed below.

The morphological characteristics, such as the chorion becoming dented and shriveled, stemmata pigmentation, and thorax and abdomen segmentation during the egg development resemble those in the dipteran insects, such as *Chrysomya megacephala* [29] and *Hermetia illucens* [30], and the hemipteran insect *Empoasca onukii* [31]. The hatching process of *P. xylostella* is very similar to that of *M. sexta*. However, the pigmentation of the stemmata and head capsule during the egg development is not mentioned in *M. sexta* [32].

The morphological characteristics, such as “head capsule slip” (HCS) and the pigmentation of the mandibles are ubiquitous in those L-L ecdyses of lepidopterans. However, the HCS in *P. xylostella* cannot be observed directly like those in *M. sexta* and *B. mori* since *P. xylostella* is much smaller in body size. We use the stemmata as a reference to the position of head capsule during the HCS according to those described in *T. ni* [9], *H. virescens* [7], and *P. includens* [8]. The ecdysis behaviors are almost identical among *M. sexta*, *B. mori*, and *P. xylostella* L-L ecdysis, but the pre-ecdysis behaviors are different. The pre-ecdysis I and II of *M. sexta* [38,39] and *B. mori* [6] perform dorsal-ventral and then posterior-ventral contractions of the abdominal segments, while it is anterior-posterior contractions in pre-ecdysis I and dorsal-ventral contractions in pre-ecdysis II in *P. xylostella,* which was not mentioned in other lepidopteran species. The wandering, HCS, and relative quiescence stages of L1–L2 ecdysis of *P. xylostella* take less time than L3–L4 of other species, but more time than L2–L3; the reason for this is not clear.

In the L-P metamorphosis, the white bars among the stemmata on the head capsule of fourth instar larvae were observed, similar to that in *H. virescens* [7]. During the pre-ecdysis period, rhythmic dorsal-ventral flexions in the anterior abdominal region named the “bowing” movement were also observed, similar to that in *M. sexta* [38]. *M. sexta* turns ventral-side-up during the ecdysis [42], while *P. includens* [8], *T. ni* [9], and *P. xylostella* keep ventral-side-down in L-P metamorphosis.

The tanning process in the pupal development is similar to *M. sexta* and *B. mori* [40]. The resorption of molting fluid and the elongation and distension of the body before eclosion of *P. xylostella* are also reported in *A. pernyi* [16]. During the eclosion process, the violent ‘shrugging’ of the wing bases is observed in *P. xylostella*, similar to *H. cecropia* [11]. The rapid rotation of the abdomen takes place sporadically in the pupal development of *H. cecropia* but is not found in the eclosion of *P. xylostella* [11].

After eclosion, *P. xylostella* can expand its wing at any surface; a hanging position is not necessary. Some species, for example, *H. zea*, need to cling to a villous surface in a hanging position for wing expansion [14], and *A. aurantiaca* climb upwards to a certain place before wing expansion [13]. *P. xylostella* holds its wings erectly after horizontal expansion without holding rigidly in a horizontal position, just like that in *M. sexta* [5], while *D. abietella* first holds its wings rigidly in a horizontal position after expansion, then vertically over the abdomen [12]. Moreover, the antennal movement after eclosion reported here were not mentioned before in other lepidopteran species.

In summary, our current findings demonstrate the morphological characters and the behavioral sequences of the molting process of *P. xylostella* for the first time. The detailed and quantified descriptions will increase our understanding of the molting process in non-model insects and facilitate the study of the underlying mechanisms.

## Figures and Tables

**Figure 1 insects-13-00289-f001:**
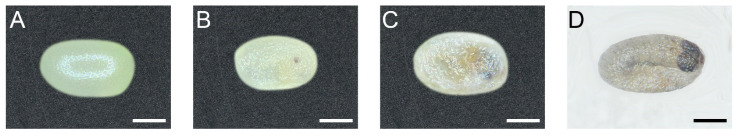
The egg development and hatching process of *P. xylostella*. Scale bar = 200 μm. *n* = 33. (**A**) Stage I: Eggs were evenly white when they were just laid, and the egg chorion was full. (**B**) Stage II: Stemmata pigmentation. Pigmentation of the stemmata began with a red spot and then darkened and, finally, turned black. (**C**) Stage III: Head capsule pigmentation. The segmentation was complete at the moment. (**D**) Stage IV: The cuticular structures of the embryo have acquired their distinctive larval morphology, and the liquid in the eggs was fully absorbed.

**Figure 2 insects-13-00289-f002:**
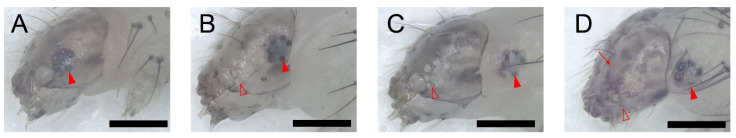
The larval–larval ecdysis of *P. xylostella* larva. The full red arrowheads point to the stemmata, and the hollow red ones point to the hollow stemmata on the old head capsule. The arrow points to the pigmented mandibles. Scale bar = 300 μm. *n* = 33 in L1–L2 ecdysis, *n* = 38 in L2–L3 ecdysis, and *n* = 53 in L3–L4 ecdysis. (**A**) Stage I: No significant change in its stemmata. (**B**) Stage II: Head capsule slip. Six stemmata on the old head capsule turned white and six “black spots” (stemmata pigment spots on the new head capsule) moved posteriorly, which means the head capsule moved anteriorly to the larva. (**C**) Stage III: Relative quiescence. All stemmata on the new head capsule have already emerged from the posterior side of the old head capsule in this stage. (**D**) Stage IV: Pigmentation of new mandibles.

**Figure 3 insects-13-00289-f003:**
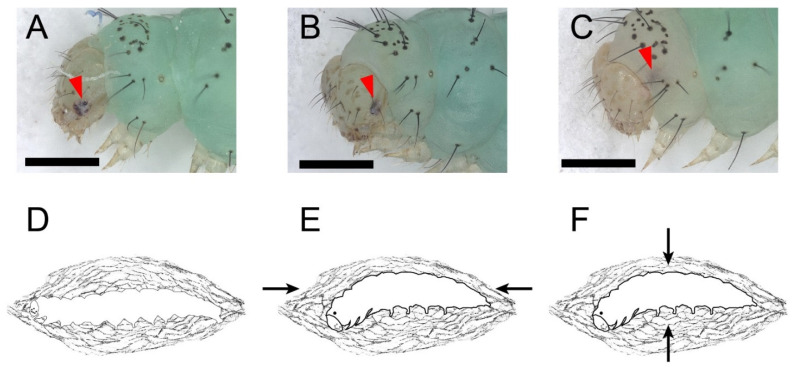
The larval–pupal metamorphosis and the tanning before the eclosion of *P. xylostella*. The full red arrowheads point to the stemmata pigment spots. Scale bar = 500 μm. *n* = 36. (**A**,**D**) Stage II: cocooning. There were a white bars among the stemmata on the head capsules of fourth instar larvae. There were no significant changes in the stemmata and the white bar at this stage. (**B**) Stage III: Head capsule slip. The head capsule slipped anteriorly and ventrally after cocooning. The stemmata pigments and the white bar progressively faded. (**C**) Stage IV: Stemmata moved out of the margin of the head capsule, merged as one pigment spot, and distinctly faded while the white bar was left inside the head capsule and almost vanished. (**E**) Pre-ecdysis I at stage IV. The arrows show the anterior-posterior contractions of the prepupa. (**F**) Pre-ecdysis II at stage IV. The arrows show the dorsal-ventral contractions of the prepupa.

**Figure 4 insects-13-00289-f004:**
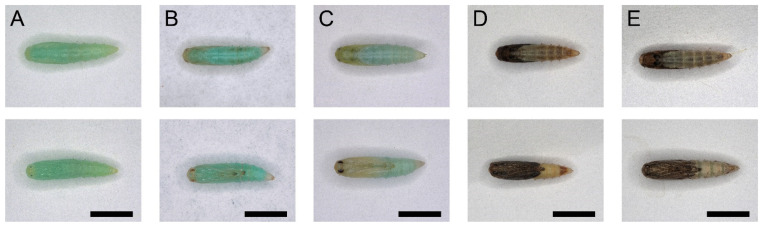
The dorsal and ventral view of the pupa development of *P. xylostella*. Scale bar = 1000 μm. *n* = 36. (**A**–**D**) Stage I of the pupa development of *P. xylostella*. (**A**) The pupa was evenly green at the end of L-P metamorphosis. (**B**) The compound eyes, antennae, and appendages turned yellow distally. (**C**) The compound eyes turned black. Meanwhile, the antennae, appendages, wings, and thorax turned yellow. The last two segments of the abdomen turned yellow with other segments kept green. (**D**) The antennae, appendages, wings, and thorax turned black. The last two segments of the abdomen turned brown, and the other segments turned yellow. (**E**) Stage II of the pupa development of *P. xylostella*: Intersegmental membrane of the pupa inflated with the metallic reflection on the scale of the pharate adult because of the full absorption of the molting fluid.

**Figure 5 insects-13-00289-f005:**
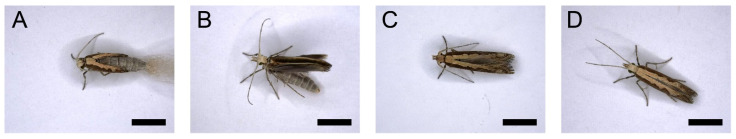
The post-eclosion behavior of *P. xylostella* moth. Scale bar = 500 μm. *n* = 36. (**A**) Stage I: The newly molted adult remained stationary. (**B**) Stage II: Wing expansion. The wings erected vertically over the abdomen. (**C**) Stage III: The wings folded downward and covered the dorsum of the adult. Meanwhile, the antennae suddenly waved backward. (**D**) Stage IV: The antennae finally waved forward.

**Table 1 insects-13-00289-t001:** Timetable of each stage of the molting process during the different developmental stages of *Plutella xylostella*. The durations are presented as means ± s.e.m. *n* = 33 in egg to first instar larva and first to second instar larva. *n* = 38 in second to third instar larva. *n* = 53 in third to fourth instar larva. *n* = 36 in fourth instar larva to pupa, pupa to adult, and post-eclosion behavior of adult. # indicates the data from Liu et al. (2002) [36].

Process	Stage	Duration
Egg-First instar larva	I	36 ± 0.2 h
	II	27 ± 0.3 h
	III	3 ± 0.1 h
	V	3 ± 0.2 h
	Hatching	0.2 ± 0.02 h
	Total developmental duration of eggs	69 ± 0.3 h
First instar larva-Second instar larva	I	91 ± 4 min
	II	72 ± 3 min
	III	378 ± 6 min
	IV	64 ± 3 min
	V	3 ± 0.1 min
	Total developmental duration of first instar larvae #	2.0 ± 0.05 d
Second instar larva-Third instar larva	I	86 ± 4 min
	II	82 ± 2 min
	III	342 ± 4 min
	IV	76 ± 2 min
	V	3 ± 0.1 min
	Total developmental duration of second instar larva #	2.2 ± 0.08 d
Third instar larva-Fourth instar larva	I	90 ± 4 min
	II	96 ± 2 min
	III	438 ± 4 min
	IV	114 ± 3 min
	V	4 ± 0.4 min
	Total developmental duration of third instar larvae #	1.5 ± 0.04 d
Fourth instar larva-Pupa	I	172 ± 14 min
	II	313 ± 11 min
	III	217 ± 9 min
	IV	459 ± 5 min
	V	4 ± 0.4 min
	Total developmental duration of fourth instar larvae #	2.0 ± 0.04 d
Pupa-Adult	I	90 ± 1 h
	II	62 ± 1 min
	III	3 ± 0.2 min
	Total developmental duration of pupae	91 ± 1 h
Post-eclosion behavior of adult	I	3 ± 0.2 min
	II	7 ± 0.2 min
	III	7 ± 0.3 min
	IV	10 ± 0.6 min

## Data Availability

The data presented in this study are available on request from the corresponding author.

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
