# Peer review of "Characterization of Molting Process during the Different Developmental Stages of the Diamondback Moth Plutella xylostella"

_insects, 2022, doi:10.3390/insects13030289_

Round 1

Reviewer 1 Report

This is an interesting study that merits publication. I only have very few minor comments.

In the aims paragraph at the end of the introduction the authors can address how this work is different from all the other studies that have been published on the subject, underlining the novelty.

How this work can be linked with the practical application of insecticides, especially those that have some ovicidal effect?

321-2. Which are the aspects that have not been reported before? 1-2 sentences can be sufficient in this paragraph.

My overall main concern is that the data are mostly presented in a rather descriptive way, with no statistics. However, I can’t see how the statistics can be incorporated here, in a sense of synthesizing morphological observations.

Reviewer 2 Report

This study focuses on detailed description of the morphological transformation and behavioral sequence of different molting processes of diamondback Moth Plutella xylostella. The authors described from the embryo to the adult the duration and the main morphological characteristics of each stages complementing previous studies. Finally, the authors provided details of the movements that precede the ecdysis.  

In general, the study contributes to the understanding of molting and ecdysis, being of interest for entomologists. In addition, this information would be useful to properly stage this insect and analyze in detail phenotypes and effects of chemicals.

I only have minor points that could help to improve the Ms.

In table 1, the different stages of ecdysis and their duration are reported. However, ecdysis only takes place at the end of each phase, so adding the total duration of each stage would help the reader to better interpret the data in the context of total complete development.

Figure 1, the background of panels A-C impede the proper visualization of the egg. I recommend a black background to better recognize the white egg.

Finally, the pictures should show the animals in each panel, anterior left and posterior right. Thus, orientation of the animals in Figure 1, 3, 4 and 5 should be flipped.

Reviewer 3 Report

I read and reviewed the manuscript titled "Characterization of Molting Process during the Different Developmental Stages of the Diamondback Moth Plutella xylostella." It provides a very detailed observation and analysis of L-L ecdysis and L-P metamorphosis, as well as eclosion behaviors. I have no major criticisms of the manuscript, with the possible exception of the organization of the figures.

Round 2

Reviewer 1 Report

The authors have addressed adequately the comments, so the paper can be accepted for publication.

Reviewer 2 Report

All the main issues have been addressed properly and now the Ms is ready for publication.

Reviewer 3 Report

Improved !